# Do Neural Transformers Learn Human-Defined Concepts? An Extensive Study in Source Code Processing Domain †

Claudio Ferretti * and Martina Saletta *

Department of Informatics, Systems and Communication, University of Milano-Bicocca, 20126 Milano, Italy
* Correspondence: claudio.ferretti@unimib.it (C.F.); martina.saletta@unimib.it (M.S.)
† This paper is an extended version of our paper published in GECCO 22: Genetic and Evolutionary Computation Conference, Companion Volume, titled *Towards the Evolutionary Assessment of Neural Transformers Trained on Source Code*.

**Abstract:** State-of-the-art neural networks build an internal model of the training data, tailored to a given classification task. The study of such a model is of interest, and therefore, research on explainable artificial intelligence (XAI) aims at investigating if, in the internal states of a network, it is possible to identify rules that associate data to their corresponding classification. This work moves toward XAI research on neural networks trained in the classification of source code snippets, in the specific domain of cybersecurity. In this context, typically, textual instances have firstly to be encoded with non-invertible transformation into numerical vectors to feed the models, and this limits the applicability of known XAI methods based on the differentiation of neural signals with respect to real valued instances. In this work, we start from the known TCAV method, designed to study the human understandable concepts that emerge in the internal layers of a neural network, and we adapt it to transformers architectures trained in solving source code classification problems. We first determine domain-specific concepts (e.g., the presence of given patterns in the source code), and for each concept, we train support vector classifiers to separate points in the vector activation spaces that represent input instances with the concept from those without the concept. Then, we study if the presence (or the absence) of such concepts affects the decision process of the neural network. Finally, we discuss about how our approach contributes to general XAI goals and we suggest specific applications in the source code analysis field.

**Keywords:** cybersecurity; static analysis; explainable AI; deep neural networks

## 1. Introduction

The recent years have been characterized by a widening diffusion of artificial neural networks (ANNs) for solving varied tasks in many different domains. With this growing popularity, and in view of neural architectures becoming more sophisticated, the research area known as explainable artificial intelligence (XAI) has gained relevance, especially when attempting to define approaches for investigating how machine learning (ML) models make their predictions.

In this work, we move in this direction by studying the internal behavior of a deep neural source code analyzer (namely, the CuBERT transformer [1]) trained in the detection of software vulnerabilities. Our goals are conceived along two lines: that of XAI, in which we investigate if (and, possibly, where) human-understandable concepts emerge in the internal activations of a complex neural network, and that of cybersecurity, in which we probe the classification of the network on the basis of emerging concepts related to the vulnerabilities over which the network is trained on so as to identify possible misconceptions or blind spots in what has been learned.

*Contributions and Outline*

The main focus of this work is to adapt the original TCAV approach [2] to fit with transformer architectures and to be applicable in the domain of source code processing. To this end, this paper extends the work presented in [3], by generalizing the approach to whole feedforward layers instead of single neurons, and by performing more detailed analysis (e.g., by considering all the 24 internal feedforward layers) and further investigations, such as the numerical assessment of the sensitivity of the network to given concepts (Table 1) and the study on how the classification accuracy of the network changes when tested on evolved sets of instances where a ground truth is imposed, and the presence of a concept is strong or weak (Table 2). In details, the contributions are the following:

- The validation of the approach on the state-of-the-art CuBERT [1] transformer, fine-tuned in the detection of software vulnerabilities.
- The definition of a fitness function for a (grammar-based) evolutionary algorithm defined as the distance from an hyperplane that, according to the presence of a given concept, separates input instances when seen as points in a space defined over the neural activations.
- A study on how the presence (or the absence) of the emerging concepts affects the decision process of the network on its original task.

In the rest of this paper, Section 2 presents a literature overview of the involved topics; Section 3 describes our analysis approach, wherein we combine the search of human-understandable concepts in the internal activations of a neural network with evolutionary techniques for assessing the classification ability; Section 4 details the performed experiments, along with technical specifics; the results are presented and discussed in Sections 5 and 6; and final remarks and possible future directions are addressed in Section 7.

**Table 1.** Conceptual sensitivity of the models to different emerging concepts.

|  | CWE 789 | CWE 369 |
|---|---|---|
| Cast to integer | 1.0 | 0.95 |
| Square brackets | 0.74 | 0.84 |
| Cyclomatic complexity | 0.58 | 0.82 |
| Many-to-many | 0.26 | 0.15 |
| *Random* | 0.49 | 0.51 |

**Table 2.** Sensitivity of the models to different strength of concepts: each cell corresponds to one concept/CWE/class experiment, and each shows classification accuracy on instances with strong presence of the concept (top triangle) and on inputs with little presence of it (bottom triangle).

| CWE | Class | Cast | # sq.br. | cycl.compl. | "I/O" |
|---|---|---|---|---|---|
| 789 | vuln. | 1.0 / 1.0 | 1.0 / 1.0 | 1.0 / 1.0 | 1.0 / 1.0 |
| 789 | safe | 1.0 / 0.8 | 0.9 / 0.8 | 0.9 / 0.8 | 0.5 / 0.7 |
| 369 | vuln. | 0.2 / ~0 | 0.2 / ~0 | 0.3 / 0.3 | 0.3 / 0.1 |
| 369 | safe | 0.3 / 0.6 | 0.4 / 0.6 | 0.7 / 0.7 | 0.2 / 0.5 |

## 2. Related Work

This work describes the experimental investigation performed for studying the decision process of a neural transformer by finding out which human-understandable concepts are the most relevant for its final prediction. This section provides a focused literature overview of the many involved topics, with the attempt of providing an overview of the state of the art of the points that are useful for our discussion.

### 2.1. ML for Static Analysis

In recent years, besides the traditional techniques and tools for static analysis [4], also approaches based on machine learning [5], and especially on deep learning [6], are becoming popular to deal with source code. Many of these techniques are derived from NLP approaches. In particular, architectures based on the transformer model [7], besides the native applicability in NLP, have proven to be effective also in the context of source code analysis: for instance, we can mention PLBART [8], a sequence-to-sequence model used for tasks such as code generation and code summarization, and CuBERT [1], a derivation of the popular BERT model [9] adjusted for processing source code instead of natural language. Among the reasons for the popularity of these kinds of neural architectures, we can mention the fact that data need only a simple preprocessing since the source code is tokenized and given as input as a vector of integers, where each integer represents a token and, most importantly, the training is performed in two steps: a first unsupervised (and computationally heavy) step in which the transformer is trained on large corpora of data, and a second supervised step, to specialize the model in a given task.

### 2.2. Evolutionary Program Synthesis

Among the evolutionary algorithms, genetic programming (GP) [10] is particularly suitable when dealing with programs since it allows to evolve individuals whose genotypes represent syntax trees. Referring to real programming languages, that are typically defined by means of a context free grammar (CFG), it would be convenient to directly work on their grammar. To this end, the technique known as grammatical evolution (GE) [11] is beneficial since, similar to GP, it is designed for evolving programs but it differs from it regarding the genotypes representation, which in GE comprises sequences of integers that can be decoded into phenotypes that are compliant with a given formal grammar.

For its inherent properties, GE has been applied, despite some limitations in solving general tasks [12], for addressing the challenging problem of program synthesis: for instance, in [13], the authors proposed a study on the importance of having a deep knowledge of the problem to solve when working with GE since this helps in designing a grammar that provides productions involving features and primitives that fit with the problem to solve, while in [14], GE is applied for synthesizing programs aiming to solve the classical problem of integer sorting.

Despite its potential in solving program synthesis tasks, the genotype representation of GE shows some defects [15]: mainly, it suffers from redundancy (i.e., several genotypes are decoded into the same phenotype) and locality (i.e., a small perturbation on the genotype leads to great changes in the corresponding phenotype). To this end, an adjustment of GE is proposed: structured grammatical evolution (SGE) [16] and its dynamic implementation (DSGE) [15], which does not need to preprocess the grammar, addresses these deficiencies with a different mapping between the genotype and phenotype.

### 2.3. Concept-Based Explainability

Essentially, in classic supervised learning scenarios, the performance of an ML model is measured by counting the ratio among the correct predictions and the wrong predictions, and by deriving scores, such as accuracy, precision and recall. However, such metrics cannot always give sufficient information about the quality of a model [17] since they only indicate the number of times that a model gives a correct or a wrong answer, but they do not give any insight of what is important for a network to make its prediction. To this end, the literature presents several works aiming at identifying which features are important for a network to make its decision [18–20]. The basic idea of such approaches is to compute the gradient of the output with respect to the input features. This is suitable for domains such as that of image processing [21,22], but in other fields (e.g., natural language processing (NLP)), where the object to analyze is a point in a discrete space and it needs a transformation to be used as input for a neural model, it is less effective. The main problem is that, in general, such a transformation from the object to a numeric input vector

is not invertible, and thus operating directly on the input features is not convenient. In the literature, we find an approach to this problem when instances are graphs in [23].

In this regard, other works operate on *concepts* rather than on features. The insight is to identify which are the human-understandable concepts that mostly influence the decision process of the network. For instance, in a past work [24], internal neurons of an autoencoder were used as classifiers for patterns not seen during the training phase. More recently, concept activation vectors (CAVs) [2] have proven to be effective to model the internal states of neural networks by means of human-understandable concepts, and similar techniques have been successfully applied in completely different domains, such as those of chess [25] and of software vulnerability detection [3].

## 3. Proposed Approach

The proposed approach is devised for studying the decision process of a neural classifier. Our experimental workflow, which is outlined in Figure 1, basically consists in treating the input instances of a neural network as points of the geometric space defined in the internal layers of the network, as detailed in Section 3.2. Together, within the input dataset, we select samples of instances that represent a given concept. In general, whatever is expressible in natural language can be considered as a concept. Here, in the specific domain of source code processing, and being that the considered dataset is composed by Java methods, we will consider both lexical concepts (e.g., the presence of targeted patterns or constructs), and syntactical ones, such as the complexity or the relation between the types of arguments and the returned objects. We can thus find an hyperplane in the activations space that separates (with some approximation) the points that represent a concept from those that do not represent it. With this equipment, we finally can consider the perpendicular direction with respect to the hyperplane as the direction representing the concept, and eventually use the signed distance from the hyperplane as a fitness function for an evolutionary algorithm to evolve programs representing a given concept.

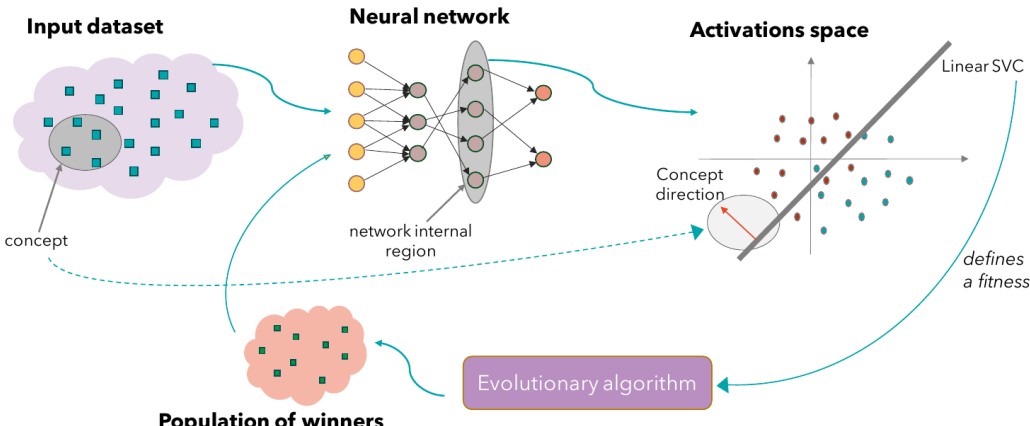

**Figure 1.** Approach overview.

### 3.1. Input Instances and Sub-Concepts

One of the crucial problems in explaining the decisions of ML models is that they usually operate on input features (e.g., matrices of pixels or sequences of textual tokens) that are difficult to be interpreted when examined by a human being. To this end, in this work, instead of operating directly on features [18], we move the focus at an higher level and reason on concepts that are meaningful for humans.

The deep neural model considered in this work, namely the CuBERT transformer [1], that will be more closely described in Section 4.1, is internally composed by 24 encoding blocks, each having a final feedforward layer. Thus, for each $l = 1, \ldots, 24$, the output of the feedforward layer $l$ can be expressed as a function $f_l : \mathbb{R}^n \to \mathbb{R}^{n \times m}$, where $n$ is the dimension of the input sequences accepted by the model, and $m$ is the number of neurons

in the layer. Notice that, as specified in this notation, each neuron has a vectorial output (rather than a scalar one) with $n$ dimensions, that is the same dimensionality of the input vector. In this paper, we will always consider feedforward layers with $m = 1024$ neurons, and input dimension $n = 512$.

In this work, the main goal is to study which human-comprehensible sub-concepts (with respect to the original task the model is trained on) are relevant for the decision process of the network, and to investigate if such emerging sub-concepts are similar to those considered by a human being when addressing the same task. For instance, if the main task is the detection of zebras in images, a possible sub-concept can be the presence of stripes in the image being tested. Similarly, in the source code analysis domain, a sub-concept related to the detection of a vulnerability such as the classic buffer overflow could be the presence of calls to unsafe library functions, such as `strcpy()`.

### 3.2. Activations Space, Linear SVCs and Concept-Based Neural Fitness Function (Content Rephrased for Similarity)

More precisely, given a sub-concept $c$, and a set $S$ of possible input instances, we can express $c$ by partitioning $S$ into two subsets $P_c$ and $N_c$ containing, respectively, the positive and negative instances with respect to concept $c$. In [2], concept activation vectors (CAVs) were used to explore the ability of layers of the network when employed to separate input instances belonging to $P_c$ or to $N_c$. Here, we define a similar approach: we consider the vector space obtained by concatenating the activations of all the neurons of a layer, and we train a linear support vector classifier (SVC) [26] to correctly recognize the activations yielded by the instances belonging to $P_c$ and those belonging to $N_c$. The rationality in choosing SVCs and linear kernels as simple concept classifiers, working on the layer activations generated from specific classes of input instances, is inspired by that found in a previous work [27], which compares activations of DNN to signals from the brain, when stimulated with some sensorial input. Formally, given a feedforward layer $l$, for each $s \in S$, we compute $f_l(s) \in \mathbb{R}^{n \times m}$, and we flatten its rows by applying a transformation $f_l^* \colon \mathbb{R}^{n \times m} \to \mathbb{R}^t$, where $t = nm$ and $f_l(s)_{x,y} = f_l^*(s)_{nx+y-1}$, for all $x \in \{1, \dots, n\}$ and $y \in \{1, \dots, m\}$. Then, we define the binary ground truth for the concept $c$ by means of the two sets $S_1 = \{f_l^*(s) \colon s \in P_c\}$ and $S_0 = \{f_l^*(s) \colon s \in N_c\}$ and then we train a linear SVC on those activation vectors. This binary classifier $V_{c,l}$, corresponding to a separating hyperplane for the activations space of layer $l$, can then be used to determine the direction of the concept $c$. By definition, a linear SVC $V_{c,l}$ separates the instances with respect to the sub-concept $c$, by means of a decision function $d_{c,l} \colon \mathbb{R}^t \to \mathbb{R}$ that represents the signed distance between the hyperplane $V_{c,l}$ and the point determined by the activations yielded by the instances in the layer $l$. Specifically, since we are considering binary classification, the SVC predicts an instance $s$ to be in class 0 or in class 1 depending on whether $d_{c,l}(f_l^*(s))$ assumes a negative or a positive value. In this way, an instance $s$ is predicted to represent or not to represent a sub-concept $c$ according to where, in the space, the activations generated in layer $l$ are placed with respect to the separating hyperplane $V_{c,l}$.

Given this premise, we define as the fitness function for our evolutionary algorithm the signed distance $d_{c,l}(s)$ computed by feeding the network with the individual $s$ and by considering the SVC $V_{c,l}$ and the corresponding point in the activations space of the feedforward layer $l$. Since high-level programming languages usually build source code programs under the control of a context-free grammar, within the scope of this paper, the idea is to apply a grammar-based genetic algorithm in order to evolve programs according to a fitness function that represents the direction of a concept. To this end, DSGE [15] is the easy choice since it allows to efficiently evolve individuals that comply with a given formal grammar. As shown in Figure 2, which reports an example of the decoding process of genotype into a phenotype on the basis of a simple BNF grammar, in DSGE, the genotypes are represented by sequences $G_1, \dots, G_n$, each listing the derivation steps of a grammar. At each position $k$, $G_k$ is referred to a non-terminal symbol of the given context-free grammar, and it is made of a list of integers whose length represents the number of

times the $k$-th non-terminal is expanded, while the values represent, for each expansion, the index in the grammar of the applied rule for that non-terminal. We remark that in DSGE, the evolution operates on genotypes; in all our experiments, at each generation, the genotypes are decoded into the corresponding phenotypes, and the fitness is computed on them.

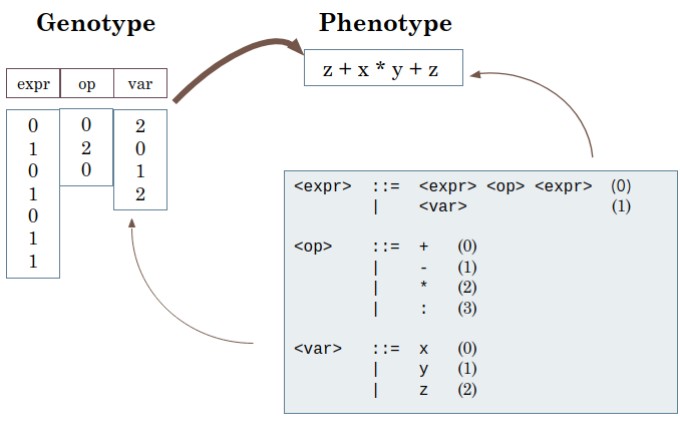

**Figure 2.** Example of DSGE decoding of a genotype into the corresponding phenotype.

*3.3. Sensitivity to Sub-Concepts*

With the outlined setting, we can now measure how much the presence (or the absence) of a concept affects the decision of the network on the task it is trained on. To this end, we addressed two approaches:

1. The first is similar to the original TCAV [2], which uses directional derivatives to compute the conceptual sensitivity on entire classes of inputs;
2. The second is based on the evolutionary synthesis of classes of inputs and on a subsequent test of the classification performance on these evolved sets.

While the first point is basically used here as a benchmark, the second is expected to be more expressive and capable of widely exploring the input space and to look for possible blind spots or misconceptions.

For each concept $c$, we consider the layer $l$ for which the support vector $V_{c,l}$ obtains the highest accuracy, and we use it for computing the fitness function of a DSGE algorithm. We operate along two directions:

- By using a simplified Java grammar (Figure 3) we generate sets of individuals that **maximize** the fitness, and on these sets we count the number of instances that are classified as vulnerable.
- By using a grammar that forces the presence of the vulnerability that the model is trained on, we generate sets of individuals that **minimize** the fitness, and on these sets, we count the number of instances that are classified as vulnerable.

Although the two points are similar in their design, they are conceived to address different issues. The maximization experiments are intended to find out the concepts that positively affect the classification. Here, the ground truth of all the evolved programs is assumed to be negative since the grammar used always produces programs that are syntactically correct but semantically meaningless, and furthermore not necessarily compilable or executable. In the minimization experiments, instead, the ground truth is assumed to be positive since, even though the grammar suffers from the same problems, it introduces lines of code that represent the vulnerability. These experiments are intended to detect which concepts lead to a misclassification when they are counteracted.

```
<start> ::= public <retype> <funid>(){\n<statements><return-stmt>;}
          | public <retype> <funid>(<parameters>){\n<statements><return-stmt>;}

<parameters> ::= <parameter>
              | <parameter>, <parameter>
              | <parameter>, <parameter>, ::= int <varid> | int[] <varid>

<statements> ::= <statement>;\n | <statement>;\n<statements>

<if-stmt> ::= if(<condition>) {\n<statements>}
           |if(<condition>) {\n<statements>} else {\n<statements>}

<condition> ::= <atom> == <atom> | <atom> != <atom> | true | false

<while-stmt> ::= while(<condition>) {\n<statements>}
              | do {\n<statements>} while(<condition>);

<digit> ::= 1 | 2 | 3 | 4 | 5 | 6 | 7 | 8 | 9        <retype> ::= int | int[] | void

<int> ::= 0 | <digit> | <digit><int>                <varid> ::= x | y | z

<return-stmt> ::= return | return <simpl-expr>       <funid> ::= funid | good | bad

<simpl-expr> ::= <atom> | <atom> <operator> <atom>    <sigint> ::= <int> | (-<int>)

<atom> ::= <varid> | <sigint> | <varid> [<int>]       <operator> ::= + | -

<statement> ::= <varinit>                 <varinit> ::= int <varid>
             | <varid>=<simpl-expr>                 | int[] <varid>
             | <varid>[<int>]=<simpl-expr>           | int <varid> = <simpl-expr>
             | (int) <varid>=<simpl-expr>            | int[] <varid> = new int[<int>]
             | <if-stmt>
             | <while-stmt>
```

**Figure 3.** Simplified Java grammar that allows the emergence of the considered possible sub-concepts.

## 4. Experiments

In the following, we describe our experiments, which essentially consist in the investigation of possible sub-concepts that emerge in the internal layers of a deep neural network trained in the classification of Java source code, and then in the application of an evolutionary algorithm able to evolve Java programs using a fitness function derived from these concepts, with the twofold objective of studying which elements mostly influence the decision of the network, and how such elements can be eventually used to deceive its prediction.

### 4.1. The CuBERT Transformer (Content Rephrased for Similarity)

In our experiments, the model we chose to analyze was the CuBERT [1] transformer, which is derived from the popular BERT [9] transformer for NLP, designed to effectively deal with source code. Transformers are becoming increasingly popular and powerful, for more and more applications different from the original NLP tasks they have been designed for (e.g., image [28] and source code [8] processing), and when using such a kind of deep architecture, the benefit is that they can be trained once on large corpora of data associated to generic and self-supervised tasks, and later they can be fine-tuned on more specialized supervised tasks. This allows to use them with high flexibility. In this work, we appointed the cybersecurity domain and fine-tuning to be on the detection of software vulnerabilities by using a CuBERT model pre-trained on a Java corpus (Available: https://github.com/google-research/google-research/tree/master/cubert, version of 11 July 2021).

To this end, we used the Juliet Test Suite v1.3 for Java [29,30], a dataset developed by the NSA that contains a corpus of programs labeled according to the common weakness enumeration (CWE) (https://cwe.mitre.org/index.html, accessed on 22 September 2022) list. For each CWE, the dataset consists of a set of Java files defining both flawed methods and, correspondingly, a number of non-flawed constructs. To build our training sets, we preprocessed the files labeled CWE-369 and CWE-789 representing the "divide by zero" and "memory allocation with excessive size value" vulnerabilities, respectively. The set of positive instances was populated with the methods that self-contain the flaw; in fact, we removed from the dataset all the methods that, to exhibit the vulnerability, need to

call other methods or to use support classes. Moreover, the set excluded all the methods whose tokenization yields vectors having more than 512 dimensions, the maximum input dimension for the CuBERT model, so as to avoid truncation of the input instances. Finally, we considered for the set of potential positive instances all the non-flawed methods labeled with any CWE, except CWE-369 and CWE-789, from the Juliet dataset.

### 4.2. Sub-Concepts Formulation

For our experiments, we considered both sub-concepts that are expected to be relevant for the detection of the considered vulnerabilities (e.g., the presence of a cast to integer), and generic sub-concepts that can be identified in the source code, but that are supposed not to be related to the vulnerabilities (e.g., the cyclomatic complexity). In details, we considered the following sub-concepts:

**Cast to integer** The presence (or absence) of a cast-to-integer operation. This sub-concept can be significant when dealing with both the divide-by-zero and uncontrolled memory-allocation vulnerabilities.

**Square brackets** The presence of an high number (i.e., >=12) of square brackets. This concept can be relevant in general since it is strictly related to the presence of an high number of accesses to array elements.

**Cyclomatic complexity** This sub-concept [31] addresses the structural complexity of a program, and it is a classical software engineering metric. It is defined as the number of linearly independent paths in the control flow graph of a program and, dealing with Java programs, can be easily computed by counting 1 point for the beginning of the method, 1 point for each conditional construct and for each case or default block in a switch-case statement, 1 point for each iterative structure and 1 point for each Boolean condition. We consider, as a sub-concept, a cyclomatic complexity higher than 10.

**I/O relationship** This sub-concept considers the *semantic* of a method in terms of the relation between what is passed as argument (i.e., the input) and the returned object (i.e., the output). We only consider a subset of all the possible I/O relations, namely, the presence or the absence of an array among the input arguments and whether the returned object is an array or a single element. In particular, in our experiments, we consider as a sub-concept the many-to-many relation, namely the methods that contain (at least) an array among their arguments, and that return an array.

**Random** This concept is defined by simply assigning random labels to the methods in the dataset. The obtained partition is obviously meaningless, and the experiments on this concept are used as a baseline to assess the validity of the other results.

Excluding the baseline random concept, the first two sub-concepts resemble some traits that a human expert takes into account when attempting to identify the presence of vulnerabilities, while the others represent elements that are relevant and recognizable, but that are not directly related to the considered vulnerabilities. As an analogy for the image-processing domain, for a model trained in recognizing zebras, we can think of concepts such as the presence of stripes or having four paws as sub-concepts related to the task, and the prevalence of a given color as an unrelated concept.

### 4.3. SVCs and Activations Spaces (Content Rephrased for Similarity)

As outlined in Figure 4, we considered the feedforward layers of the encoder blocks, namely the dense layers that feed the succeeding encoder block. In the CuBERT transformer, there are 24 encoding blocks, each one composed by 1024 neurons with a 512-dimensional output. For each considered sub-concept, we started by collecting the activations obtained by flattening the output of the feedforward layers (see Section 3.2) for a balanced random sample, where 100 program instances were associated to the concept, and 100 did not represent it. Then, for both the fine-tuned models (i.e., the binary classifiers trained in

the detection of the divide by zero and uncontrolled memory allocation vulnerabilities, respectively), and for each concept, we gathered the activations of the flattened layers as points of a $512 * 1024$-dimensional vector space, and for each of the 24 layers, we fit a linear support vector classifier (SVC) to correctly separate the points representing programs related to the chosen concept, and the points representing programs with no presence of the same concept.

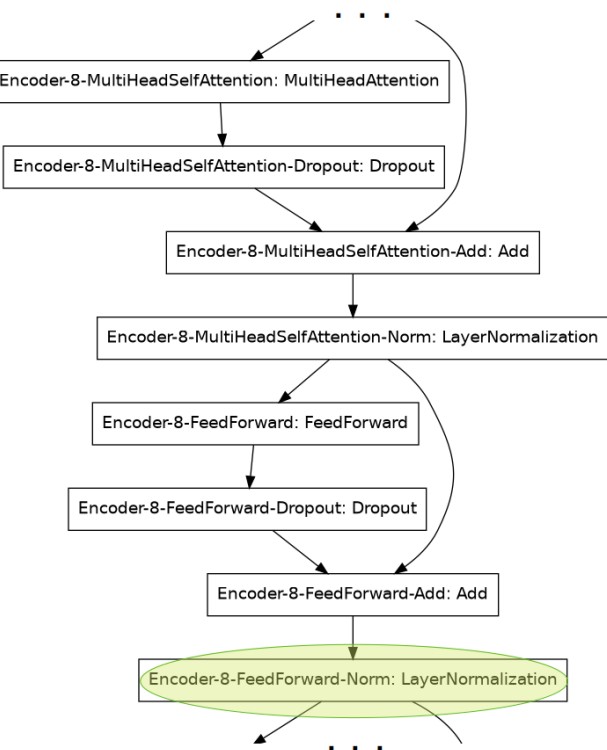

**Figure 4.** An encoder block of the CuBERT transformer. The activation spaces are defined on the highlighted feedforward layers.

Ad the end of this procedure, besides the average accuracies obtained by the SVC over 10 runs, which are reported in Figures 5 and 6, for each concept, we identified the "best" layer (i.e., the layer that can most fruitfully used to distinguish programs that hold the concept from programs that do not, by considering its activation values) and the corresponding best SVM. The latter was chosen as a fitness evaluator for an evolutionary algorithm, as detailed in Section 4.4.

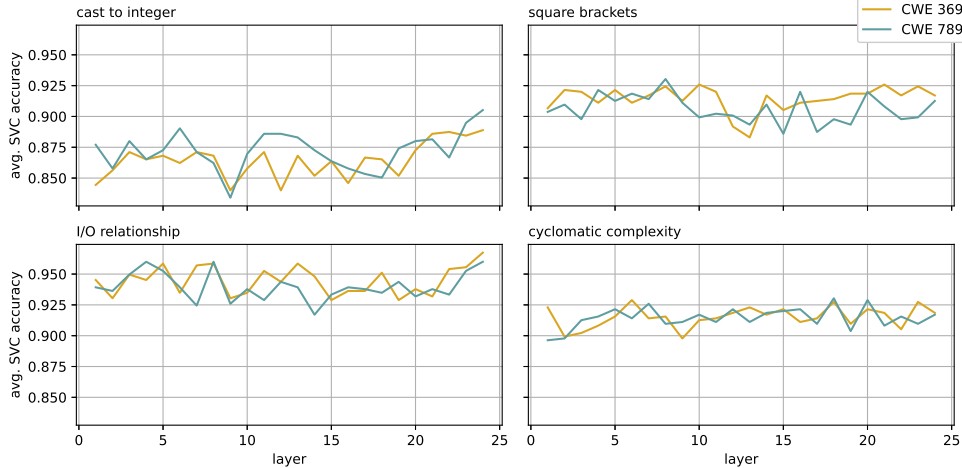

**Figure 5.** Average accuracies obtained over 10 runs by the linear SVCs trained on different concepts.

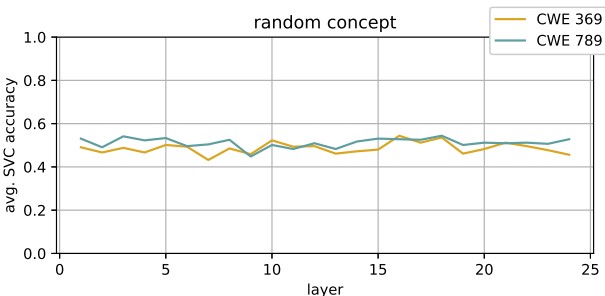

**Figure 6.** Average accuracies obtained over 10 runs by the linear SVCs trained on random concepts.

*4.4. Evolutionary Search along Sub-Concepts Directions*

Given the trained neural networks, in our case two transformers, fine-tuned respectively to recognize CWE-369 and CWE-789, we analyze each with respect to how they are influenced in their accuracy when instances have strong or weak presence of a given concept so as to discover possible biases in the way they classify them. Our method builds source code instances with DSGE on a Java grammar, which is simplified but flexible enough to produce the syntactical elements required by the specific concepts, and by the specific classification tasks. Moreover, in our search, we can also force the grammar to always generate individuals which contain vulnerabilities for the CWE we are considering. This is obtained by modifying the grammar with the addition of production rules expanding into vulnerable constructs which the DSGE cannot rule out.

The DSGE fitness function in our experiments is exactly the decision function of a chosen SVC, which has been built, as previously seen, associated to a concept and to a trained network layer. The decision function takes as input the activation vector of the layer, activations generated by the network when it is fed with the source code of an individual of the evolving population. Among the ones that we built for several layers in our networks, we chose the SVCs with good accuracy for the concepts they represent. In this way, the fitness values are higher when the presence of the concept is stronger in the input source code since the SVC decision function value is related to the distance of the input activation vector from the separating hyperplane that the trained SVC uses to perform the concept classification task.

Each evolutionary run is defined by the following:

- The considered concept, among the five we described in Section 4.2;
- The considered CWE, between CWE-369 and CWE-789;
- Whether we define a grammar which forces the corresponding vulnerability or not;
- Whether we maximize or minimize the fitness value.

For a set of combinations of these choices, DSGE is run for 12 generations, with a population of 200 individuals and the elitism of 4 individuals, with probability equal to 0.1 for mutation and 0.9 for crossover.

In this way, we search for Java methods which maximize or minimize the presence of a concept in terms of the signal inside the network. These instances usually do not belong to the datasets used during training, and we will later check how the networks classify them, taking into account the possible presence of a vulnerability forced by our grammar.

*4.5. Measuring Sensitivity to Sub-Concepts*

We finally studied how much the prediction of the model is influenced by the concepts by performing two analyses. The first one, which is derived from the TCAV approach [2], basically measures the percentage of instances that represent a concept among a set of programs that are positive with respect to a classification task. Formally, for each concept $c$, we considered a sample $X_v$ of 100 methods from the dataset that are both labeled and predicted to be vulnerable by our model. Notice that we considered the vulnerable class, but the same considerations can be made on the other class of safe programs. Then, we

measured the sensitivity $S_{c,v}$ by computing, $\forall x \in X_v$, the distance $d_{c,l}(f_l^*(x))$, where $l$ is the layer referred to the best support vector $V_{c,l}$ trained on the concept $c$, and by computing the ratio between the number of times that $d_{c,l}(f_l^*(x))$ has a positive value, and the cardinality of sample $X_v$:

$$S_{c,v} = \frac{|\{x \in X_v : d_{c,l}(f_l^*(x)) > 0\}|}{|X_v|} \tag{1}$$

A second analysis was performed on the instances generated via evolutionary search. We measured how accurately a given network classifies the instances generated by evolutionary means in our experiments, where we chose one of our 4 concepts, a single CWE, and a configuration among the following two:

- Instances that are not vulnerable, and whose fitness value is maximized (the presence of the chosen concept is strong);
- Instances that are vulnerable, and whose fitness value is minimized (the presence of the chosen concept is weak).

This last choice aims at checking the impact of a strong concept when the network should classify the instance as safe, and conversely in the case of instances known to be vulnerable. We recall that among the four concepts we identified, two are commonly considered relevant when manually looking for CWE-369 or CWE-789 vulnerabilities, while two are higher-level concepts, which we identified as instances of what we can probe for in our classifiers.

This leads to 16 different experiments, and from each, we eventually obtain a population of source code methods, evolved according to the chosen combination of CWE, concept, and desired presence of the concept. For each experiment, we ask of the network, trained on the chosen CWE, to classify all the individuals generated during the 12 generations of the run, taken without repetitions. We do this separately on two subsets of the total population of an experiment: the half population with the individuals of highest fitness values, and the half with the lowest fitness values. Finally, the relevance for a network of a sub-concept is measured by looking at the absolute accuracy of the classification on each of the two subsets, and by analyzing the relative accuracy between them. This will allow to study how a concept impacts classification, both on safe and on vulnerable source code.

## 5. Results

The results of the first experimental phase, in which we looked for the emergence of the mentioned concepts, are reported in Figures 5 and 6. We trained, for each concept and for 10 runs, a linear SVC in separating the activations yielded by all the 24 layers on balanced samples of 2000 functions, where 1000 represent the concept and the other 1000 do not represent it. It can be observed that the two models exhibit a similar pattern for all the considered concepts, but that different concepts define problems of different difficulties: for instance, the average accuracy for the cyclomatic complexity concept is around 91% in all the layers, while for the cast to integer we move from 80% to 90%. Some sub-concepts (i.e., cyclomatic complexity) have a constant average accuracy along the layers, while others (e.g., cast to integer) present some increasing or decreasing trends. Note also that, as expected, the accuracy of the SVCs trained on random concepts (Figure 6) is always around 50%, thus confirming the validity of the results obtained on the well-defined concepts.

After having observed the *absolute* emergence of the concepts, we measured their impact on the original task. We performed two analyses. The results of the first one, obtained by considering only vulnerable instances and that are directly derived from the original TCAV [2], are reported in Table 1, and they give interesting insights on how different concepts have a different influence on the decision. For instance, it is clear that the presence of a cast to integer has a much stronger positive effect than the many-to-many I/O relationship. Additionally, notice that the influence of the randomly defined concept is irrelevant, since for both the models, it is about 50%.

For the second measure of sensitivity we propose, based on the classification of instances which are generated by the DSGE evolutionary algorithm, we performed 16 experiments, each corresponding to a combination of choices for concept, CWE, and maximization or minimization of fitness, respectively, on safe and on vulnerable code. As a result, each experiment generated an overall population of around 2200 unique individuals, whose fitness ranges were polarized toward positive or negative values by the chosen optimization. Nonetheless we checked the values ranges, and they always included both positive and negative values, apart from experiments aiming at minimizing for CWE-789 the presence of "cast" elements in the input, where no individual with negative fitness value was generated (bottom-left box in Figure 7). We recall that having fitness values spanning both positive and negative values means that each experiment generated both individuals with the strong presence of a concept together with individuals with little presence of the same concept.

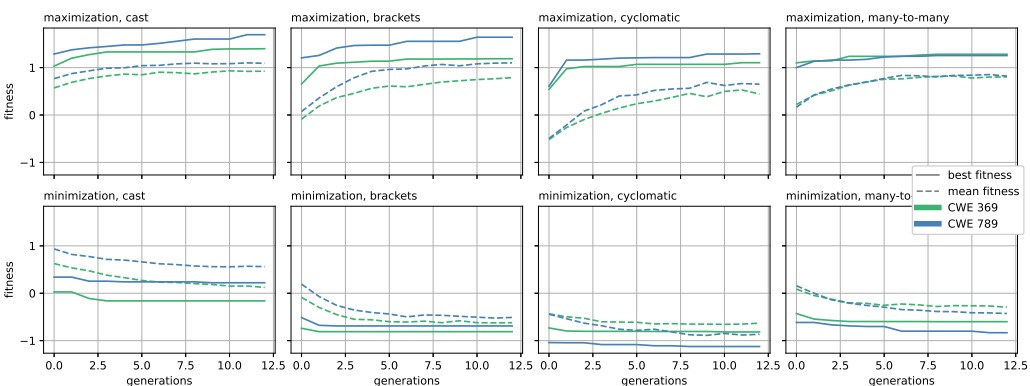

**Figure 7.** Fitness obtained when generating input instances with DSGE, maximizing or minimizing the considered concepts. Notice that, in the minimization experiments, the grammar is modified in order to include a vulnerability, while the ground truth of the individual evolved when maximizing the fitness is assumed to be negative.

The accuracy of our fine-tuned transformers on each (sub)population is presented in Table 2. Each cell corresponds to a given choice for concept, CWE, and ground truth of individuals (we maximized the concept for safe instances, i.e., with ground truth 0, and minimized it for vulnerable ones, i.e., with ground truth 1). In each cell, the upper part reports the classifier accuracy on instances with strong presence of the concept, and conversely, the lower part shows the accuracy on those with little concept signal.

The numbers show quantitatively how input with more, or less, signal for a concept (i.e., the value of the decision function of the corresponding SVC, on activations induced by the instance) affects the accuracy of the classification for safe and for vulnerable inputs. As we can see, the two networks have a different behavior on the evolved instances, and are also different with respect to each concept.

For instance, we can state that the classifier for CWE-789, when fed with the instances we generated outside the training set, has better accuracy, and it seems less impacted by the presence, or absence, of our concepts. On the other hand, the transformer for CWE-369 has generally worse accuracy, on such instances. It will be interesting to analyze its accuracy values, such as those resulting on vulnerable instances with many square brackets, where variations in their presence changes accuracy from around 0 to 0.2. The next section discusses these outcomes.

## 6. Discussion

The previous sections introduced the goals, the approach, and the raw outcomes of the experiments. Quantitative results allow analysis of the behavior of the neural classifiers. We can discuss the outcomes of evolutionary exploration of classifiers' behavior, whose outcomes are presented in Table 2. We aim at discovering mistakes of the neural networks, and possible reasons for them, after the training ended with high accuracies

over the training/test datasets. According to our approach, the experiments first required that we identified understandable concepts that could affect the classification, and then those concepts allowed to train and select a corresponding SVC on a whole neural layer. Finally, the best SVC provided a fitness function to control the evolutionary search of input instances, either strongly or weakly related to the concept.

The numbers that tell us something about the neural classifiers that we are checking are all in Table 2. Overall, each row in the table gives us information about the performance of a classifier on the positive and negative sets we generated. In our case, the classifier trained for CWE-789 performed much better than the one for CWE-369. Moreover, the latter shows better performance on safe instances with little presence of our concepts (the bottom half-cells of the bottom row in the table). These numbers also are evidence that we obtained ample sets of adversarial instances, among the ones we generated, and suggest which features in source code are more effective in deceiving the classifiers.

From data in the table, we can move to finer observations. For instance, the CWE-369 classifier's accuracy seems immune to effects from the presence of high cyclomatic complexity in the input instances. Instead, for all the other concepts, it appears that their presence increases the probability for the classifier to consider instances as vulnerable since for instances known as vulnerable, the accuracy increases with more signal and conversely it decreases for safe input instances, with wider or smaller variations for different concepts. The only anomaly we found for the performance of the CWE-789 classifier can be extrapolated from the worst accuracy in the second row of the table. It appears that when input methods have an array among their arguments and return an array, i.e., the "I/O" concept is strong, accuracy drops to 0.5, and this would deserve further investigation.

As a final remark, we stress that these results will be relevant depending on the choice of human concepts. Such a choice has to be made manually and with a good knowledge of the application domain, security of source code in our case. However, sometimes, given the complex statistical basis behind the internal reasoning of neural networks, even probing them with instances associated to concepts which seemingly are unrelated to the domain, could uncover unexpected mistakes in their classifying behavior.

## 7. Conclusions and Further Directions

We developed and tested a procedure to probe a neural classifier with respect to instances having some human-defined characteristics, that we called sub-concepts. The results are promising since for the neural networks we analyzed, we were able to extract insights on the their internal reasoning from quantitative results of our experiments.

This work could extended by considering different source code analysis tasks and, correspondingly, different concepts. Finally, this analysis could be improved so as to look for the emerging of concepts with different abstraction levels along the different layers of networks, from the ones closer to the input to the deepest ones.

**Author Contributions:** Conceptualization, C.F. and M.S.; Methodology, C.F. and M.S.; Investigation, C.F. and M.S.; Writing—original draft, C.F. and M.S.; Writing—review & editing, C.F. and M.S. All authors have read and agreed to the published version of the manuscript.

**Funding:** This research received no external funding.

**Institutional Review Board Statement:** Not applicable.

**Informed Consent Statement:** Not applicable.

**Data Availability Statement:** The pre-trained CuBERT model is publicly available in the Google Research GitHub repository: https://github.com/google-research/google-research/tree/master/cubert, access on 22 September 2022. Juliet Test Suite for Java used for fine-tuning CuBERT and for the sub-concepts investigations is developed by the NSA Center for Assured Software and can be downloaded from https://samate.nist.gov/SARD/test-suites/111, access on 22 September 2022. The DSGE library used for the evolutionary experiments is available on GitHub: https://github.com/nunolourenco/sge3, access on 22 September 2022.

**Conflicts of Interest:** The authors declare no conflict of interest.

**Abbreviations**

The following abbreviations are used in this manuscript:

| | |
|---|---|
| ANN | Artificial Neural Network |
| BNF | Backus–Naur Form |
| CAV | Concept Activation Vector |
| CFG | Context Free Grammar |
| CWE | Common Weakness Enumeration |
| DSGE | Dynamic Structured Grammatical Evolution |
| GE | Grammatical Evolution |
| GP | Genetic Programming |
| ML | Machine Learning |
| NLP | Natural Language Processing |
| SVC | Support Vector Classifier |
| XAI | Explainable Artificial Intelligence |

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
