# Peer review of "Do Neural Transformers Learn Human-Defined Concepts? An Extensive Study in Source Code Processing Domainâ€"

_algorithms, doi:10.3390/a15120449_

Round 1
Reviewer 1 Report
The article is devoted to the development and application of artificial intelligence methods. The topic is undoubtedly relevant.
There are a number of questions about your article.
1. You are using linear support vector classifier. What is the basis of the hypothesis of linear separability?
2. If you used kernels in the linear support vector classifier algorithm, which ones? Has the choice of kernel been examined for the result obtained?
3. Increase the size of the figure 5
Author Response
Thank you for your comments and suggestions.
1. You are using linear support vector classifier. What is the basis of the hypothesis of linear separability?
We do not aim at a good classifying performance with SVCs against the layers we measure. Support vector classifiers are used to check how much they are enough to work downstream from the layers, which take instances and generate activations, to separate instances related or not to a concept.
The approach is inspired by the one taken in literature when comparing activations of DNN to signals from the brain, when stimulated with some sensorial input, as in REF: "Schrimpf, Martin, et al. 'Brain-score: Which artificial neural network for object recognition is most brain-like?.' BioRxiv (2020): 407007", and we added the reference and some text in Section 3.2 to make this explicit.
2. If you used kernels in the linear support vector classifier algorithm, which ones? Has the choice of kernel been examined for the result obtained?
Mostly for the reasons described for the first remark, we built support vector classifiers with just linear kernels.
3. Increase the size of the figure 5
agreed, for readability, we enlarged the text present in the figure.
We did the same also for figure 6.
Reviewer 2 Report
In this work, the authors investigate the challenges arising from explainable artificial intelligence in the context of non-invertible transformations. The work is framed as an experimental investigation, with several open questions spanning different topics (including neural network explainability, cybersecurity, and evaluation design).
The topic of this paper is relevant, and the manuscript includes experimental results on a source code transformer (cuBERT).
However, I think that there are several major issues in the manuscript in its current form:
- The authors should better clarify the main contribution of this work. The abstract and the introduction (and the title, too) introduce many concepts, but it is not clear what is the main contribution and novelty. I suggest changing the title and abstract, better clarifying the main contribution, and restructuring the rest of the manuscript to emphasize the main story, keeping more exploratory results and discussion in specific sections.
- Similarly, the authors should clarify what are the main results, and distinguish them from discussion/exploration points. I suggest adding a section at the end of the Introduction, clarifying not only the main differences with respect to the conference paper (the authors already did that), but also the global contributions of the manuscript in its current form, clearly distinguishing between methodological improvements (new methods, refined methods), experimental results, exploration analysis, and discussion points.
- I suggest reviewing the related work section and expanding the "concept-based explainability" subsection. There are several statements without citations, which do not appear to be motivated. For example, lines 106-108: "but in other fields, where the object to analyse needs a transformation to be used as input for a neural model, it is less effective.". I'm not aware of such limitations for discrete data, and there are many methods tailored to discrete data (sequences, graphs, etc.).
- The experiments should include more baselines and ablation studies, to identify which parts of the proposed method matter more. Also, as previously mentioned, the authors should better clarify the purpose of these experiments, clearly explaining how each experiment relates to each question in the introduction.
Author Response
Thank you for your comments and suggestions.
- The authors should better clarify the main contribution of this work. The abstract and the introduction (and the title, too) introduce many concepts, but it is not clear what is the main contribution and novelty. I suggest changing the title and abstract, better clarifying the main contribution, and restructuring the rest of the manuscript to emphasize the main story, keeping more exploratory results and discussion in specific sections.
We modified the title and the abstract according to this remark. Now we introduce as key enabling contribution the definition of the TCAV-like approach to study if human understandable concepts emerge in the internal layers of a sophisticated neural network (namely, the CuBERT transformer) and if they affect the decision wrt the original task. All the performed experiments (e.g. SVCs on layers activations, evolutionary exploration of the input-output relation for the transformer) are to assess the validity of such hypothesis.
- Similarly, the authors should clarify what are the main results, and distinguish them from discussion/exploration points. I suggest adding a section at the end of the Introduction, clarifying not only the main differences with respect to the conference paper (the authors already did that), but also the global contributions of the manuscript in its current form, clearly distinguishing between methodological improvements (new methods, refined methods), experimental results, exploration analysis, and discussion points.
Along the same lines chosen for title and abstract, we changed the Introduction to answer this remark.
- I suggest reviewing the related work section and expanding the "concept-based explainability" subsection. There are several statements without citations, which do not appear to be motivated. For example, lines 106-108: "but in other fields, where the object to analyse needs a transformation to be used as input for a neural model, it is less effective.". I'm not aware of such limitations for discrete data, and there are many methods tailored to discrete data (sequences, graphs, etc.).
We added to the related work part a (hopefully) better explanation of the point and a supporting reference from literature.
- The experiments should include more baselines and ablation studies, to identify which parts of the proposed method matter more. Also, as previously mentioned, the authors should better clarify the purpose of these experiments, clearly explaining how each experiment relates to each question in the introduction.
We performed, and presented, an added experiment with respect to a "random" concept in order to better validate our analysis on the results with meaningful concepts. We did not consider ablation studies, mostly because all our measures are related to whole layers, a component whose availability cannot be cleanly constrained in the model's architecture. On the other hand, Table 2 summarizes our final experiments giving figures for all the opposing setups with respect to the impact of having, or not having, any of the concepts we chose in the instances.
Round 2
Reviewer 1 Report
The authors have corrected the comments. Accept the article.